# On What Tasks Did Children between the Ages of 3 and 12 Years Spend Their Time during the COVID-19 Pandemic? An International Comparative Study between Ibero-America and Europe

**DOI:** 10.3390/children9070971

**Published:** 2022-06-29

**Authors:** Alberto Sanmiguel-Rodríguez, Mª Luisa Zagalaz-Sánchez, Víctor Arufe-Giráldez, Javier Cachón-Zagalaz, Gabriel González-Valero

**Affiliations:** 1Faculty of Language and Education, University of Camilo José Cela, 28692 Madrid, Spain; 2Department of Didactics of Musical, Plastic and Body Expression, Faculty of Humanities and Educational Sciences, University of Jaén, 23071 Jaén, Spain; lzagalaz@ujaen.es (M.L.Z.-S.); jcachon@ujaen.es (J.C.-Z.); ggvalero@ujaen.es (G.G.-V.); 3Specific Didactics Department, Research and Diagnostic Methods in Education, Education Faculty, University of a Coruña, 15001 Coruña, Spain; v.arufe@udc.es

**Keywords:** psychosocial well-being, sociodemographic factors, physical activity, educational level of the parents, COVID-19

## Abstract

The pandemic caused by COVID-19 meant, in many countries, the establishment of a period of confinement in which families were forced to restrict movement and social contacts with the consequent risk of inactivity. Our objective as to analyze the degree of psychosocial well-being, sociodemographic aspects and use of technological means depending on the educational level of the parents. The sample consisted of 2316 children aged between 3 and 12 years (M = 7.70; SD = 2.86). For the analysis and treatment of the data, the statistical software SPSS 25.0 (IBM Corp, Armonk, NY, USA) was used. We found that the European participants used more video consoles (M = 0.89 ± 1.33) and tablets (M = 1.30 ± 0.95), while the Ibero-Americans obtained higher values in the use of TV (M = 2.28 ± 1.10) and levels higher in a negative state of psychosocial well-being (M = 7.29 ± 1.07) and in tiredness/fatigue (M = 4.34 ± 2.44). We concluded that, during the period of confinement in European areas, higher values were obtained in the time dedicated to Physical Activity (PA), use of tablets, school task performance, artistic activities, family games, reading, free play and hours of sleep; while in Ibero-America, there were longer times in the use of technological devices and performing domestic tasks.

## 1. Introduction

The global spread of SARS-CoV-2 (COVID-19) led the World Health Organization (WHO) to declare a pandemic in March 2020. To slow down this transmission, many countries took strict socio-sanitary measures that have affected the lifestyles of millions of people and children, not being able to continue carrying out activities like the ones they did until confinement [1]. Thus, the pandemic caused by COVID-19 led to the establishment of a period of confinement, and families were forced to restrict movement and social contacts with the consequent risk of inactivity [2,3]. More than 90% of children were affected by the restriction of movement to curb the transmission of the new coronavirus that culminated in the closure of all educational and social activities [4], which significantly affected families [5]. 

Parental and family factors influence many aspects of their children’s lives, including their level of Physical Activity (PA) [6]. The COVID-19 lockdown measures resulted in children and adolescents staying and learning at home [7]. With schools closed, the frequency with which parents conducted literacy activities at home with their children and the sociodemographic variables that influence this collaboration are unknown [8].

Confinement can have short- and long-term impacts on people’s mental health and quality of life. Knowing what factors cause stress can benefit the development of strategies and resources for future situations [9,10]. The effects of COVID-19 containment measures on children’s emotional and behavioral development are not sufficiently clear [11]. Therefore, the negative impact of confinement on the emotions and behavioral aspects of children highlights the need for strategic approaches, especially for those most susceptible due to environmental factors and pre-existing emotional problems [11].

Approximately 35.1% of the parents reported that the psychological health of their children was considerably affected. The most important concern was social isolation [12], as well as unemployment, the increase in family conflicts, the lack of opportunities for teleworking and the deterioration of the psychological health of the parents that can cause deterioration in the conditions physical and mental health of children and adolescents [12].

Since COVID-19 dramatically changed human social life, restrictive lockdown periods to curb the spread of the virus were found to particularly affect the psychological well-being of children and their families [13]. For its part, technological media are present in many homes establishing changes in social communication and lifestyle [14]; however, the COVID-19 pandemic has further highlighted the deep digital divide and the enormous challenge its education system faces in continuing to teach students during the lockdown [15].

Due to the above, this study was proposed to verify the influence of the educational level of the parents on the behavior of children aged 3 to 12 years during the COVID-19 pandemic, both in Ibero-American and European countries. The objective was to analyze the tasks to which children between 3 and 12 years of age dedicated their time.

## 2. Materials and Methods

### 2.1. Study Design and Participants

For this study, a descriptive, comparative, cross-sectional design was used, with a single measurement for a single group. This study was conducted by the Universities of Jaén, Coruña and Granada. The sample consisted of 2316 children aged between 3 and 12 years (M = 7.70; SD = 2.86). The sampling was for convenience, inviting families with children in the Early Childhood and Primary education stage to participate during the COVID-19 pandemic. Regarding gender, the distribution of the sample was homogeneous, representing 52.4% boys (*n* = 1214) and 47.6% girls (*n* = 1102). Specifically, 54.7% (*n* = 1268) of the sample represented Europe and 45.3% (*n* = 1048) Ibero-America. However, Table 1 provides the distribution according to the countries that participated.

### 2.2. Variables and Instruments

A self-prepared questionnaire (ad-hoc) was used to record the sex and age of the children, the origin of the responses (Europe and Ibero-America) and the level of education of the parents (categorized into “basic education”, “medium education”, “higher studies” and “postgraduate”). The questionnaire on Equipment and Use of Information and Communication Technologies in Households (TIC-H2019) prepared by the National Institute of Statistics (INE) following the recommendations of the Statistical Office of the European Union (EUROSTAT) was used to know the means and technological resources that families had at home during confinement, as well as the time of use, expressed in minutes, of children under 12 years of age. 

Due to the influence of lifestyle on indoor activities during confinement, parents indicated the time expressed in minutes that their children spent on daily activities such as PA practice, household chores, playing instruments, artistic activities, household chores, playing with the family, reading and playing freely. For the degree of psychosocial well-being, a Likert-type scale with 10 response options was used (where 1 = “null” and 10 = “extreme maximum”). The degree of happiness, energy, tiredness/fatigue, self-esteem and creativity during confinement was recorded, allowing the sum of the state in general to be established, as well as the mean values of a degree of positive and negative psychosocial well-being.

### 2.3. Procedure

Through the Google Forms platform, a questionnaire called “Boys, girls and confinement” was created. Electronic information was disseminated through social networks, in order to reach the population under study (relatives residing in Europe and Ibero-America with children under 12 years of age), as well as contacting various education professionals who had access to sufficient families to ensure good dissemination of the instrument. The questionnaire was activated during the various confinement periods established in Ibero-America and Europe.

Out of a total of 2598 responses, a total of 237 (9.12%) questionnaires were eliminated because they were not correctly filled in or belonged to another educational stage. Thus, the final sample consisted of 2316 children. By completing the form, all participants gave their consent to work with the data anonymously. Throughout the investigation, the ethical principles reflected in different documents and official treaties on research ethics were taken into account, thus, guaranteeing the anonymity of the participants, the confidentiality of the data reflected in the questionnaires and other ethical considerations related to the investigation research in education [16,17].

### 2.4. Analysis of Data

A descriptive analysis is conducted to determine the sociodemographic characteristics and behaviors during the pandemic, applying means (M), standard deviations (SD) and frequencies (%). Likewise, the Kolmogorov–Smirnov test was performed to determine the normality and homogeneity of the variance in the variables. To establish the differences between the variables, the Student’s *t*-test was used for independent samples and the one-factor ANOVA, using Pearson’s Chi-square statistical indicator, to establish the differences. The statistical software SPSS 25.0 (IBM Corp, Armonk, NY, USA) was used for data analysis and treatment.

## 3. Results

Table 2 establishes the parameters related to the number of technological devices in relation to the area of residence of the participants, for which statistically significant results were obtained (*p* ≤ 0.05). We found that the European participants used more video consoles (M= 0.89 ± 1.33) and tablets (M = 1.30 ± 0.95), while the Ibero-Americans were the ones who obtained higher values in the use of TVs (M= 2.28 ± 1.10).

For Table 3, the differences were recorded according to the area of residence, taking into account the time spent using technological devices and daily actions conducted during confinement. Statistically significant relationships were also obtained for these results (*p* ≤ 0.05). In European areas, higher values were manifested in the time spent performing PA (M = 36.97 ± 34.36), use of tablets (M = 30.61 ± 48.53), school task performance (M = 117.16 ± 84.05), artistic activities (M = 53.18 ± 48.32), playing with the family (M = 86.21 ± 65.55), reading (M = 29.39 ± 21.05), free play (M = 99.53 ± 77.40) and hours of sleep (M = 9.60 ±7.21). While in Ibero-America, there were longer times in the use of video consoles (M = 27.98 ± 55.74), computers (M = 47.29 ± 75.43), mobile phone (M = 47.18 ± 75.76) and performing domestic tasks (M = 25.96 ± 31.14).

In addition, the relationship between the state of psychosocial well-being and the area of residence was established (Table 4), for which significant differences of *p* ≤ 0.05 were obtained. Ibero-American participants presented higher levels for a negative state in psychosocial well-being (M = 7.29 ± 1.07) and in tiredness/fatigue (M = 4.34 ± 2.44). While the European subjects showed higher mean values for the creativity dimension (M = 7.88 ± 1.94).

Considering the place of residence, Table 5 and Table 6 analyze the number of devices according to the level of education of the parents. After establishing statistically significant relationships (*p* ≤ 0.05) for European subjects, we found that parents with basic education had a greater number of televisions (M = 2.51 ± 1.19), while parents with medium education had more video consoles (M = 1.07 ± 1.72). Finally, those who had postgraduate education reported having more computers (M = 2.48 ± 1.25) and tablets (M = 1.47 ± 1.11). However, in Ibero-America, parents with postgraduate education presented the highest mean values for all devices: televisions (M = 2.47 ± 1.18), game consoles (M= 0.98 ± 1.23), computers (M = 2.34 ± 1.15) and tablets (M = 1.15± 0.97).

Table 7 and Table 8 analyze the relationships between the time spent using technological devices and actions conducted during confinement in relation to the level of education of the parents and the area of residence. Considering the European parents, statistically significant differences were obtained (*p*≤ 0.05). Those children whose parents had a basic education level spend longer time using the mobile phone, expressed in minutes of use per day (M = 30.24 ± 59.73). As for those with an average level of education, they obtained higher values in the time spent watching television (M = 92.84 ± 64.07), development of artistic tasks (M = 60.65 ± 51.12) and domestic tasks (M = 26.02 ± 20.23). Likewise, the subjects with postgraduate education stated that their children used the computer more (M = 46.82 ± 74.68) and obtained more hours of sleep (M = 9.72 ± 1.20).

In the case of Ibero-America, statistically significant differences were also obtained (*p* ≤ 0.05). Parents with a basic level of education presented the lowest levels in all the study variables, except for the time dedicated to domestic tasks. Although, differences compared to the Europeans were perceived, since the parents with higher education presented the highest mean values in the practice of PA (M = 27.24 ± 30.02), use of the mobile phone (M = 50.02 ± 79.35), school task performance (M = 111.78 ± 83.95), time dedicated to playing musical instruments (M = 8.22 ± 19.07), reading (M = 23.08 ± 26.89) and free play (M = 87.50 ± 86.78). 

In addition, children of parents with postgraduate education spent more time using video consoles (M = 34.17 ± 60.17), television (M = 98.61 ± 77.38), computer (M = 50.99 ± 79.49) and tablets (M = 30.68 ± 54.03), as well as more time spent playing with the family (M = 57.45 ± 48.13) and hours of sleep (M = 9.00 ± 1.19). All these values refer to average values of time in minutes per day, except for sleeping hours, which are hours per day.

Considering the state of psychosocial well-being according to the level of education of the parents and the area of residence (Table 9 and Table 10), statistically significant results were obtained at the level of *p* ≤ 0.05. European children of parents with postgraduate education, obtained higher levels in the state of positive psychosocial well-being (M = 7.98 ± 1.31) and the dimensions of happiness (M = 8.12 ± 1.41), energy (M = 8.38 ± 1.68) and self-esteem (M = 8.08 ± 1.67). 

However, the subjects whose parents had a basic educational level, showed higher values for the negative state of psychosocial well-being (M = 6.13 ± 1.28) and in tiredness/fatigue (M = 4.14 ± 2.44). Considering Ibero-American children, differences were only obtained in the negative state of psychosocial well-being (M = 6.81 ± 1.53) and tiredness/fatigue (M = 5.11 ± 2.76), with the highest mean values manifesting in parents with basic educational level.

## 4. Discussion

### 4.1. PA, Socioeconomic Aspects and Psychosocial Well-Being

Regarding the analysis of sociodemographic and PA aspects, the findings of this study indicate that children residing in Europe showed higher values in the time spent doing PA, free play and hours of sleep. In the same way, the children whose parents had the highest educational level had the highest time for PA.

Following these lines, Aguilar-Farias et al. [18] showed that, during the early stages of the pandemic in Chile, time spent in PA and sleep quality decreased, while recreational screen time and sleep duration increased. Children with space to play at home and living in rural areas experienced attenuated impact of the pandemic restrictions on their PA levels, screen time and sleep quality. Older children, those whose parents were 35 to 45 years old and more educated, and those who lived in apartments had greater changes, mainly a decrease in total PA and an increase in screen time.

On the other hand, Arufe-Gíraldez et al. [2] detected levels of PA lower than those recommended, with a mean of 31.81 min compared to the 180 min recommended by the WHO regarding the practice of PA. These authors pointed out that the practice of PA was directly associated with time in front of television, computers, degree of tiredness and creativity. In another study [19], PA levels were analyzed and found to be low, as was the time spent on activities such as music or games. Likewise, Zagalaz-Sánchez et al. [1] confirmed a statistically significant influence of the conditions of the house and the place of residence in the daily time dedicated to different educational activities, such as reading, PA, free play or the use of technological devices among children who live in small flats and those who live in large flats or houses with gardens and those who live in urban and rural settings.

Gilic et al. [6] highlighted the importance of the parent–child relationship and parent/family support in promoting PA both during everyday life and during crises and challenging health situations such as the COVID−19 pandemic. For their part, Christner et al. [13] found that children aged 7–10 years had more emotional symptoms as well as fewer behavioral problems and hyperactivity than children aged 3–6 years. Children’s and their parents’ stress level, the extent to which children missed other children, and children’s age were negatively related to children’s overall life satisfaction.

Single parenthood and being an only child were associated with higher levels of childhood problems. Other authors [8] indicated that all the indicators and sociodemographic characteristics (the age of the children, the number of children in the family and the educational level of the parents) were significant, with the dialogical-creative literacy activities and the digital literacy activities performed less frequently by parents.

On the other hand, this study analyzed psychosocial well-being, and the results showed that Ibero-American participants had higher levels for a negative state, both in psychosocial well-being and in tiredness/fatigue. Mazza et al. [20] found that lower extraversion and more emotional and hyperactive-inattentive child symptoms were significant predictors of parental distress. In addition, a significant two-way interaction persists between child emotional problems and parental extraversion. In general, parents express high rates of psychological distress, indicating serious concerns during lockdown. Likewise, families with a child who suffers from emotional and behavioral difficulties must be detected immediately by social services in order to activate support interventions in order to prevent chronic and amplified manifestations of these problems.

Fasano et al. [21] indicated that the implications of confinement and social isolation measures caused by confinement by COVID-19 in children and their parents are still unknown, according to which, 5% of children showed changes in their emotional state, 55.3% altered their routine and 62.6% showed sleep disorders. Families with lower socioeconomic status were more concerned about health, food shortages and household income. For these authors, the adverse emotional state of the children was associated with the feeling of loneliness of the parents and inversely with the maintenance of a routine. The routine and positive attitude of parents benefits the well-being of children.

In other research, Westrupp et al. [22] indicated that subjective well-being levels during the pandemic were considerably lower than pre-lockdown ratings. During the pandemic, a lower subjective well-being was associated with a low educational level of the parents. Such well-being in parents raising children of these ages appears to be disproportionately affected by the COVID-19 pandemic. Specific risk groups for whom government intervention may be warranted include parents in socially disadvantaged backgrounds, parents with pre-existing mental health problems, and parents facing significant pandemic-related job changes [22]. For Bilal et al. [9], there were statistically significant differences in parental stress with respect to parental marital status, age, gender, and employment status. Such differences did not exist in parental stress regarding the number and age of children and parental education [9].

In a later study, Bourion-Bédès et al. [7] noted that difficulties in isolating at home were associated with poor health-related quality of life on the dimensions of psychological well-being, relationships with parents, and autonomy. In addition, living in a small apartment, not leaving the house and having indoor noise in the home were associated with poor health-related quality of life.

### 4.2. Educational Level of the Parents

The findings of this study indicate that those who had postgraduate education reported having more computers and tablets. European children of parents with postgraduate education obtained higher levels in the state of positive psycho-social well-being and the dimensions of happiness, energy and self-esteem.

Prior to this study, some authors had stated that there are no differences in the volume of PA according to educational level; however, there is a tendency for it to be higher. The different or equal educational level of the parents does not affect the volume of PA practice [23]. Ajejas et al. [24] showed that the prevalence of overweight and obesity in Spain was increasing, and was higher in boys than in girls. According to data from 2011, children who did not engage in any type of PA or whose parents had a low level of education showed the highest prevalence of obesity [24].

The results of Ribeiro et al. [5] revealed that Portuguese parents supported their children during the pandemic mainly through monitoring attention in class and homework completion. However, several variables seemed to significantly determine the time of parental involvement, which was greater when the students attended public schools, when they were less autonomous and younger, when the educational level of the parents was lower, when the child was a male and when online school time was greater. The findings of these authors [5] highlighted the need for a significant investment of time by parents, particularly of children in Primary Education, which makes it difficult to cohere work or teleworking with school activities. Most children spend 2–4 h a day studying, while parents help them at least half the time. Parents mainly explain homework instructions, review their children’s work and teach new topics. To a lesser extent, they help their children to solve tasks [25]. 

Following these contributions, the results of Bokayev et al. [26] showed that the age of the parents and the level of family income were positively correlated with the level of parental satisfaction with distance learning, while the number of children in a family was negatively related to the satisfaction of the parents with the learning process. Olaseni and Olaseni [4] revealed that the parents’ socioeconomic status significantly impacted forced learning in students during the pandemic lockdown. Likewise, the personal characteristics and personality of the parents impacted forced learning during confinement in Nigeria.

Girls, older youth, youth with a lower socioeconomic status and youth with a migrant background from developing countries seemed to experience the lockdown as more difficult, thus, possibly accentuating the need for services in these groups [27]. Other authors [28] suggested that children with families that established consistent bedtimes and read stories to them more frequently had better scores during the early school years, and this variable was more related than the socioeconomic status of the families and the parents’ educational level. According to Kotrla et al. [29] a lower amount of time that children spent using a tablet or smartphone for entertainment purposes and a higher educational level of parents were positively related to more frequent engagement and time spent in interactive reading with children.

### 4.3. Use of Technological Means

We found that the European participants presented more game consoles and tablets, while it was the Ibero-Americans who had a greater number of televisions. Following these lines, Cachón-Zagalaz et al. [19] indicate that the use of digital screens is an important part of the daily routine of children at home. The time children spent sleeping was directly proportional to the time spent in PA and indirectly proportional to the time spent looking at screens. The children who slept the most were those between 0 and 3 years old, especially girls who belonged to large families. PA levels in the sample were low, as were the times spent on activities, such as music or games.

Briggs [30] noted that higher-educated, high-income parents who owned a computing device and have internet access at home preferred online classes compared to low-income parents with high school education and less. On the other hand, Nagata et al. [31] indicated that children had a mean of 3.99 h of screen time per day with most of the time watching TV shows or movies (1.31 h), playing video games (1.06 h) and watching/streaming videos (1.05 h).

On average, black children use 1.58 h more screen time per day and Asian children 0.35 h less screen time per day compared to white children (median 3.46 h per day); these tendencies persisted in most of the modalities. Boys reported more total screen time (0.75 h more) than girls, which was mainly attributed to video and video games. Girls reported more time texting, social networking, and video chatting than boys.

Higher earnings were associated with lower screen time use in all modalities except video chat [31]. According to Bergmann et al. [32] parents reported that young children not required to be in school were exposed to more screen time during lockdown than before. While this was exacerbated in countries with longer lockdowns, there was no evidence that increased screen time during lockdown was associated with sociodemographic variables, such as child age and socioeconomic status. However, screen time during lockdown was negatively associated with socioeconomic status and positively associated with child age, caregiver screen time and children’s attitudes toward screen time. Serra et al. [33] indicated a more frequent use of smartphones among Italian children and adolescents during the COVID-19 pandemic, compared to the pre-lockdown period.

This may be related to the social distancing measures adopted during the months of the pandemic. Authors [33] observed a significant increase in the excessive use of technological media, which led to many situations of children with sleep disorders. The results of Martínez-Domínguez and Fierros-González [15] showed that the probability of having children with Internet access and use patterns depended on the level of schooling and economic situation of the parents, digital skills and place of residence, as well as the presence of Internet devices. These findings suggest the urgent need to redesign the current policy for the use of technological media with a comprehensive long-term vision that guarantees access to new technologies and their productive use for students immersed in an ecosystem of educational innovation for the next century.

## 5. Conclusions

We concluded that, during the period of confinement in European areas, higher values were obtained in the time dedicated to PA, use of tablets, school task performance, artistic activities, family games, reading, playing free and hours of sleep; while in Ibero-America, there were longer times in the use of technological devices and performing domestic tasks.

The findings of this research reveal that Ibero-American participants presented higher levels of negative states in psychosocial well-being and in tiredness/fatigue. European parents with basic education had a greater number of TVs, while those with medium education had more video consoles, and those with postgraduate education reported having more computers and tablets. Likewise, European children of parents with postgraduate education, obtained higher levels in the state of positive psychosocial well-being, while Ibero-American parents who had postgraduate education showed a greater number of technological devices.

## 6. Limitations and Future Prospects

A limitation of the study is that the data were analyzed through surveys conducted by parents. In future studies, it is suggested that the cases be analyzed more closely at the individual level to determine patterns of behavior of the children depending on the educational level of the parents. A comparison could also be made with other countries to see if the patterns are the same as in this research.

For all these reasons, public and educational institutions should establish social and educational policies that guarantee a minimum of information and training for parents so that they can establish strategies at home to promote healthy habits and responsible use of technological media.

## Figures and Tables

**Table 1 children-09-00971-t001:** Distribution of the study sample according to participating countries.

Country	Frequency (*n*)	Percentage (%)
Mexico	163	7.0
Paraguay	19	0.8
Colombia	28	1.2
Bolivia	21	0.9
Cuba	23	1.0
Chile	46	2.0
Perú	13	0.6
Guatemala	91	3.9
Brazil	38	1.6
Argentina	337	14.6
Ecuador	265	11.4
Costa Rica	4	0.2
France	90	3.9
Poland	28	1.2
Portugal	457	19.7
Spain	693	29.9

**Table 2 children-09-00971-t002:** Number of technological devices depending on the area of residence.

					Levene’s Test		*t* Test for Equality of Means	
Characteristic	Category		M	DT	F	Sig	t	Sig. (Bilateral)
Number of devices	Televisions	Ibero-America	2.28	1.10	44.414	0.000	5.938	0.000
		Europe	2.02	1.01				
	Game consoles	Ibero-America	0.69	0.96	3.545	0.060	−4.203	0.000
		Europe	0.89	1.33				
	Computers	Ibero-America	1.77	1.11	10.978	0.001	−1.654	0.098
		Europe	1.85	1.08				
	Tablets	Ibero-America	0.79	10.87	3.741	0.053	−13.457	0.000
		Europe	1.30	0.95				

**Table 3 children-09-00971-t003:** Time of use of technological devices and daily actions (expressed in minutes) during confinement as a function of area of residence.

					Levene’s Test		*t* Test for Equality of Means	
Characteristic	Category		M	DT	F	Sig	t	Sig. (Bilateral)
Time spent	PA	Ibero-America	24.80	28.84	0.926	0.336	−9.109	0.000
		Europe	36.97	34.36				
	Game consoles	Ibero-America	27.98	55.74	10.937	0.001	2.015	0.044
		Europe	23.65	47.73				
	Televisions	Ibero-America	84.70	77.94	75.101	0.000	−0.724	0.469
		Europe	86.74	57.86				
	Computers	Ibero-America	47.29	75.43	44.167	0.000	6.212	0.000
		Europe	29.59	61.73				
	Tablets	Ibero-America	23.88	53.81	0.002	0.962	−3.161	0.002
		Europe	30.61	48.53				
	Mobile phone	Ibero-America	47.18	75.76	206.563	0.000	11.367	0.000
		Europe	18.12	45.90				
	Homework	Ibero-America	103.22	85.39	0.906	0.341	−3.943	0.000
		Europe	117.16	84.05				
	Musical instruments	Ibero-America	6.95	18.27	0.005	0.945	−0.790	0.429
		Europe	7.59	20.18				
	Artistic activities	Ibero-America	32.77	45.06	17.492	0.000	−10.433	0.000
		Europe	53.18	48.32				
	Housework	Ibero-America	25.96	31.14	82.189	0.000	3.254	0.001
		Europe	22.49	19.79				
	Play as a family	Ibero-America	53.09	51.40	59.652	0.000	−13.623	0.000
		Europe	86.21	65.55				
	Reading	Ibero-America	20.95	24.32	28.913	0.000	−8.953	0.000
		Europe	29.39	21.05				
	Free play	Ibero-America	78.55	81.35	0.291	0.590	−6.343	0.000
		Europe	99.53	77.40				
	Hours of sleep	Ibero-America	8.85	1.18	0.718	0.397	−14.795	0.000
		Europe	9.60	1.21				

**Table 4 children-09-00971-t004:** State of psychosocial well-being according to the area of residence.

					Levene’s Test		*t* Test for Equality of Means	
Characteristic	Category		M	DT	F	Sig	t	Sig. (Bilateral)
State of psychosocial well-being	Positive	Ibero-America	6.28	1.26	0.141	0.707	1.633	0.103
		Europa	5.95	1.14				
	Negative	Ibero-America	7.29	1.07	15.406	0.000	6.721	0.000
		Europa	7.10	1.04				
	Happiness	Ibero-America	7.97	1.51	0.775	0.379	−0.340	0.734
		Europa	7.99	1.50				
	Energy	Ibero-America	8.23	1.68	4.137	0.042	0.005	0.996
		Europa	8.23	1.62				
	Tiredness and/or fatigue	Ibero-America	4.34	2.44	5.275	0.022	6.809	0.000
		Europa	3.67	2.30				
	Self-esteem	Ibero-America	8.05	1.66	0.051	0.821	1.703	0.089
		Europa	7.93	1.63				
	Creativity	Ibero-America	7.69	1.87	0.579	0.447	2.413	0.016
		Europa	7.88	1.94				

**Table 5 children-09-00971-t005:** Number of devices according to the educational level of European parents.

Characteristic	Category		M	DT	F	X^2^
Number of devices	Televisions	Basic education	2.51	1.19	23.686	0.000
		Mid-level education	2.18	1.02		
		Higher education	1.96	0.90		
		Postgraduate	1.66	1.05		
	Game consoles	Basic education	0.99	1.19	3.186	0.023
		Mid-level education	1.07	1.72		
		Higher education	0.79	1.01		
		Postgraduate	0.90	1.62		
	Computers	Basic education	1.34	0.74	47.848	0.000
		Mid-level education	1.51	0.93		
		Higher education	1.90	1.04		
		Postgraduate	2.48	1.25		
	Tablets	Basic education	1.07	0.91	4.850	0.002
		Mid-level education	1.29	0.92		
		Higher education	1.31	0.91		
		Postgraduate	1.47	1.11		

**Table 6 children-09-00971-t006:** Number of devices according to the educational level of Ibero-American parents.

Characteristic	Category		M	DT	F	X^2^
Number of devices	Televisions	Basic education	1.87	1.02	8.165	0.000
		Mid-level education	2.12	1.08		
		Higher education	2.37	1.06		
		Postgraduate	2.47	1.18		
	Game consoles	Basic education	0.41	0.84	10.477	0.000
		Mid-level education	0.53	0.74		
		Higher education	0.73	0.95		
		Postgraduate	0.98	1.23		
	Computers	Basic education	1.24	0.95	38.863	0.000
		Mid-level education	1.36	0.99		
		Higher education	1.89	1.07		
		Postgraduate	2.34	1.15		
	Tablets	Basic education	0.36	0.59	29.210	0.000
		Mid-level education	0.52	0.71		
		Higher education	0.88	0.88		
		Postgraduate	1.15	0.97		

**Table 7 children-09-00971-t007:** Time of use of technological devices and daily actions (expressed in minutes) during confinement according to the educational level of European parents.

Characteristic	Category		M	DT	F	X^2^
Time spent	PA	Basic education	35.07	28.79	1.674	0.171
		Mid-level education	33.49	39.92		
		Higher education	38.21	32.80		
		Postgraduate	39.17	34.13		
	Game consoles	Basic education	28.21	48.52	1.060	0.365
		Mid-level education	22.66	44.37		
		Higher education	22.00	44.32		
		Postgraduate	26.99	60.06		
	Televisions	Basic education	82.48	63.22	2.711	0.044
		Mid-level education	92.84	64.07		
		Higher education	83.20	50.81		
		Postgraduate	92.15	64.44		
	Computers	Basic education	18.74	34.32	7.904	0.000
		Mid-level education	23.51	46.60		
		Higher education	29.05	66.45		
		Postgraduate	46.82	74.68		
	Tablets	Basic education	37.09	62.57	1.748	0.155
		Mid-level education	28.05	41.22		
		Higher education	29.06	43.63		
		Postgraduate	34.46	59.72		
	Mobile phone	Basic education	30.24	59.73	4.830	0.002
		Mid-level education	20.75	49.52		
		Higher education	14.72	40.79		
		Postgraduate	16.82	43.59		
	Homework	Basic education	106.90	83.24	2.220	0.084
		Mid-level education	119.63	81.80		
		Higher education	114.64	83.17		
		Postgraduate	128.42	89.40		
	Musical instruments	Basic education	6.68	21.90	2.383	0.068
		Mid-level education	5.04	11.54		
		Higher education	8.78	23.60		
		Postgraduate	8.00	16.23		
	Artistic activities	Basic education	53.56	51.55	3.117	0.025
		Mid-level education	60.65	51.12		
		Higher education	51.00	46.90		
		Postgraduate	49.46	45.65		
	Housework	Basic education	22.29	21.28	5.150	0.002
		Mid-level education	26.02	20.23		
		Higher education	20.62	18.55		
		Postgraduate	23.54	21.18		
	Play as a family	Basic education	83.59	69.54	1.795	0.146
		Mid-level education	91.61	67.77		
		Higher education	87.02	68.27		
		Postgraduate	78.16	48.66		
	Reading	Basic education	29.29	25.58	0.914	0.433
		Mid-level education	27.67	20.20		
		Higher education	29.82	20.06		
		Postgraduate	30.50	21.79		
	Free play	Basic education	88.96	79.90	2.259	0.080
		Mid-level education	108.62	82.47		
		Higher education	98.34	74.94		
		Postgraduate	97.91	75.24		
	Hours of sleep	Basic education	9.38	1.37	5.258	0.001
		Mid-level education	9.45	1.16		
		Higher education	9.54	1.19		
		Postgraduate	9.72	1.20		

**Table 8 children-09-00971-t008:** Time of use of technological devices and daily actions (expressed in minutes) during confinement according to the educational level of Ibero-American parents.

Characteristic	Category		M	DT	F	X^2^
Time spent	PA	Basic education	18.49	26.07	4.520	0.004
		Mid-level education	20.82	27.49		
		Higher education	27.24	30.02		
		Postgraduate	26.81	27.59		
	Game consoles	Basic education	16.55	50.09	5.269	0.001
		Mid-level education	19.18	49.81		
		Higher education	32.45	57.41		
		Postgraduate	34.17	60.17		
	Televisions	Basic education	44.26	61.1	9.060	0.000
		Mid-level education	79.77	76.90		
		Higher education	88.37	78.92		
		Postgraduate	98.61	77.38		
	Computers	Basic education	23.70	50.98	2.919	0.033
		Mid-level education	44.65	69.90		
		Higher education	50.76	79.27		
		Postgraduate	50.99	79.49		
	Tablets	Basic education	10.00	30.01	4.859	0.002
		Mid-level education	16.96	45.28		
		Higher education	27.42	59.71		
		Postgraduate	30.68	54.03		
	Mobile phone	Basic education	24.40	57.31	3.502	0.015
		Mid-level education	48.73	78.87		
		Higher education	52.02	79.35		
		Postgraduate	39.22	62.98		
	Homework	Basic education	80.08	90.63	5.153	0.002
		Mid-level education	92.23	85.42		
		Higher education	111.78	83.95		
		Postgraduate	105.54	84.39		
	Musical instruments	Basic education	2.81	8.92	3.196	0.023
		Mid-level education	5.11	17.15		
		Higher education	8.22	19.07		
		Postgraduate	7.94	20.04		
	Artistic activities	Basic education	23.08	41.28	2.298	0.076
		Mid-level education	29.81	43.82		
		Higher education	34.19	43.02		
		Postgraduate	37.48	53.43		
	Housework	Basic education	28.31	44.21	1.166	0.321
		Mid-level education	24.44	31.97		
		Higher education	27.39	29.44		
		Postgraduate	23.22	28.07		
	Play as a family	Basic education	35.61	43.13	3.284	0.020
		Mid-level education	52.23	52.63		
		Higher education	54.51	52.41		
		Postgraduate	57.45	48.13		
	Reading	Basic education	15.72	20.53	3.958	0.008
		Mid-level education	17.96	21.56		
		Higher education	23.08	26.89		
		Postgraduate	21.74	21.10		
	Free play	Basic education	49.81	74.89	8.161	0.000
		Mid-level education	65.35	68.88		
		Higher education	87.50	86.78		
		Postgraduate	85.78	81.48		
	Hours of sleep	Basic education	8.52	1.15	3.456	0.016
		Mid-level education	8.76	1.20		
		Higher education	8.90	1.17		
		Postgraduate	9.00	1.19		

**Table 9 children-09-00971-t009:** State of psychosocial well-being according to the educational level of European parents.

Characteristic	Category		M	DT	F	X^2^
State of psychosocial well-being	Positive	Basic education	7.50	1.60	5.705	0.001
		Mid-level education	7.72	1.43		
		Higher education	7.94	1.48		
		Postgraduate	7.98	1.31		
	Negative	Basic education	6.13	1.28	2.920	0.033
		Mid-level education	5.80	1.15		
		Higher education	5.97	1.09		
		Postgraduate	5.95	1.13		
	Happiness	Basic education	7.66	1.70	5.815	0.001
		Mid-level education	7.79	1.58		
		Higher education	8.09	1.45		
		Postgraduate	8.12	1.41		
	Energy	Basic education	8.12	1.87	2.845	0.037
		Mid-level education	8.02	1.64		
		Higher education	8.30	1.52		
		Postgraduate	8.38	1.68		
	Tiredness and/or fatigue	Basic education	4.14	2.44	2.401	0.046
		Mid-level education	3.58	2.18		
		Higher education	3.66	2.32		
		Postgraduate	3.52	2.23		
	Self-esteem	Basic education	7.52	1.87	7.403	0.000
		Mid-level education	7.69	1.64		
		Higher education	8.07	1.53		
		Postgraduate	8.08	1.67		
	Creativity	Basic education	7.35	2.07	1.868	0.133
		Mid-level education	7.70	1.93		
		Higher education	7.76	1.76		
		Postgraduate	7.67	1.94		

**Table 10 children-09-00971-t010:** State of psychosocial well-being according to the educational level of Ibero-American parents.

Characteristic	Category		M	DT	F	X^2^
State of psychosocial well-being	Positive	Basic education	8.23	1.45	1.262	0.286
		Mid-level education	7.90	1.40		
		Higher education	7.93	1.39		
		Postgraduate	8.03	1.38		
	Negative	Basic education	6.81	1.53	5.505	0.001
		Mid-level education	6.29	1.19		
		Higher education	6.27	1.26		
		Postgraduate	6.09	1.20		
	Happiness	Basic education	8.10	1.72	0.375	0.771
		Mid-level education	7.97	1.51		
		Higher education	7.93	1.48		
		Postgraduate	8.03	1.54		
	Energy	Basic education	8.51	1.56	1.144	0.330
		Mid-level education	8.16	1.72		
		Higher education	8.20	1.65		
		Postgraduate	8.35	1.74		
	Tiredness and/or fatigue	Basic education	5.11	2.76	4.957	0.002
		Mid-level education	4.43	2.34		
		Higher education	4.36	2.45		
		Postgraduate	3.84	2.36		
	Self-esteem	Basic education	8.17	1.54	2.337	0.072
		Mid-level education	7.97	1.74		
		Higher education	7.98	1.65		
		Postgraduate	8.34	1.53		
	Creativity	Basic education	8.44	1.72	2.438	0.063
		Mid-level education	7.79	1.95		
		Higher education	7.90	1.91		
		Postgraduate	7.75	2.09		

## Data Availability

Not applicable.

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
