# Peer review of "On What Tasks Did Children between the Ages of 3 and 12 Years Spend Their Time during the COVID-19 Pandemic? An International Comparative Study between Ibero-America and Europe"

_children, 2022, doi:10.3390/children9070971_

Round 1

Reviewer 1 Report

Review of an article What tasks did children between the ages of 3 and 12 spend their time on during the COVID-19 pandemic? An international comparative study between Latin America and Europe

I have read the manuscript which presents results from the cross-sectional study conducted via an online Google form questionnaire which parents of children who reside in European counties and in Latin American countries fulfilled. The study analysed the degree of psychosocial well-being, sociodemographic aspects and use of technological means by children depending on the educational level of their parents during the COVID-19 lockdown. It also presented the number of screen-media uses by children and the time they spend using them during the time of the COVID-19 lockdown. The manuscript has to be improved. There are corrections within the Materials and Method section, presentation of data within Results, and construction of the Discussion section. All manuscript has to be written in the passive voice. Specific corrections and suggestions are listed below.

Title: I suggest correcting the title similar to: On what tasks did children between the ages of 3 and 12 years spend their time during the COVID-19 pandemic? An international comparative study between Latin America and Europe.

Introduction

Line 33: when first mentioning the COVID-19, use the full name, and then its abbreviation.

Line 34-35: correct “people and children”

Line 54-56: correct the sentence, either use “one third” or “35.1%”

Line 68-72: I suggest writing the study aim and/or hypothesis/es clearly. It is afterwards easier to present the results, answer them in the discussion, guide the discussion and clearly conclude.

Material and Methods

2.1. It is not clearly presented. Mention the Institution approving the study conduction. There is no information on which European and Latin American countries the participants were from. I suggest listing them, to provide information about statistically significant number calculations.

Line 91-95: this sentence is difficult to follow. I suggest simplifying it. For example: Due to lifestyle influence on indoor activities during the confinement, parents indicated the time express….

Line 108: provide a total number of responses received, and make a calculation about the responses not included in the study.

Results

This section should be written in the passive voice. I suggest correcting this issue. When presenting the statistical significance, provide the exact data about it, not mention “(p<0.05)”. The presented data in text and in the tables are not clear and not easy to follow. There is no data about the time-frequency the participants used screen-media devices, is it min per week or hour per week or per day? Please, correct this in the text and in the tables accordingly.

Line 141: correct the sentence, for example, Children residing in Ibero-America spent more time using video consoles, etc. Also, unify the term, Is it Latin America or Ibero-America?

Line 115 to 163: since there are large tables, I suggest writing the name of the table after the data presented, for easier reading and following the results.

Line 189-190: correct the sentence, I suggest similar to Those children whose parents had a basic education level spend longer time using the mobile phone, etc. Il lines afterwards write the time (h or min/week or day) because it is not clear and it is difficult to follow.

Line 198: I suggest not mentioning the European zone, I suggest using the term “compared to the Europeans” or similar. Also, state the name of the value (I suppose that is the time and then provide a unit of it). I suggest not to use “development of school tasks”, perhaps use “school task performance” or similar to this.

Line 203: correct the sentence, rather use “spent more time using video, etc. than “had a long time”. Also, mention the name of the table after the data is presented, and mention “when compared to other education level subgroups”.

Line 253-256: I suggest correcting the sentence, avoid “educational background basics” rather say as it is mentioned in the tables “basic studies” i.e. “basic educational level”.

Discussion

I suggest rewriting the discussion section again. Authors write it to firstly mention the study results, then to present the literature review about that subject, but all without the meaning of their results regarding the presented literature review. It is not clear to read, and follow and not easy to understand what the authors want. When presenting the review of one issue, it is needed to put it into the context of their results and what they really mean for the future, to discuss its meaning. It is missing the connection to all those literature cites that the authors thoroughly presented. Also, I suggest shortening, and to connect that literature review with their results, in a way to comment on them are they similar or higher or larger or longer or smaller, shorter, etc. to the mentioned. I.e. to compare the study results with existing literature analysed.

Specifically:

Line 273: correct the sentence, it is confusing, those are the children residing in Europe, not the parents who spent their time doing PA the most when compared to the children from Latin America.

Line 275: correct the sentence, those are not the parents, these are the children whose parents had the highest educational level to have the highest time for physical activity. 

Line 398: correct this sentence, I suggest rephrasing it, not to mention “children in colour” despite the original literature mentioning this way. Be ethical in this issue.

Line 421: this sentence is not clear, is this the meaning of the authors or of the literature authors. Also, correct its original meaning.

There is no study limitations and strengths. I suggest that the authors provide one.

Conclusion

The conclusion is needed to rewrite again and directed more to the significant meaning for the future studies and paediatric actions.

Line 441: there is not clear which one are the reasons, who are they, please, name it, be specific about those reasons and their future implications.

Tables

The tables heading are not clear, they should be specific and precise about what is presented. It lacks the parameter's value (time or the score), the number of participants, and their origin place. The tables presented are too long, they could be merged into one according to the same subject. The table could present the mean and SD and a p-value of each parameter in one row and the columns can merge the same subject but for each subgroup (Europe and Latin America). The tables also lack subheadings about which statistical test is applied.

Author Response

After attending to all the comments made by the reviewers, we believe that the work has substantially improved in quality.

Here are the changes made based on your feedback:

Comment

Title: I suggest correcting the title similar to: On what tasks did children between the ages of 3 and 12 years spend their time during the COVID-19 pandemic? An international comparative study between Latin America and Europe.

Response

It has changed.

Comment

Line 33: when first mentioning the COVID-19, use the full name, and then its abbreviation.

Response

it has been done as indicated

Comment

Line 34-35: correct “people and children”

Response

It has changed.

Comment

Line 54-56: correct the sentence, either use “one third” or “35.1%”.

Response

It has changed.

Comment

Line 68-72: I suggest writing the study aim and/or hypothesis/es clearly. It is afterwards easier to present the results, answer them in the discussion, guide the discussion and clearly conclude.

Response

It has changed.

Comment

2.1. It is not clearly presented. Mention the Institution approving the study conduction. There is no information on which European and Latin American countries the participants were from. I suggest listing them, to provide information about statistically significant number calculations.

Response

Thank you very much for your comment. We have included the universities that conducted the study, as well as a list of the countries participating in the study.

Comment

Line 91-95: this sentence is difficult to follow. I suggest simplifying it. For example: Due to lifestyle influence on indoor activities during the confinement, parents indicated the time express….

Line 108: provide a total number of responses received, and make a calculation about the responses not included in the study.

Response

Thank you very much for your comment. The changes have been made, the sentences have been reworded and the total number of questionnaires deleted has been calculated.

Comment

This section should be written in the passive voice. I suggest correcting this issue.

Response

It has been observed that in this section there are texts in the passive voice.

Comment

When presenting the statistical significance, provide the exact data about it, not mention “(p<0.05)”. The presented data in text and in the tables are not clear and not easy to follow. There is no data about the time-frequency the participants used screen-media devices, is it min per week or hour per week or per day? Please, correct this in the text and in the tables accordingly.

Response

Thank you very much for your comment. With regard to providing specific significance, we consider that taking into account the premises of the journal and the other reviews, it is more interesting to indicate that there are statistically significant differences (in a general way) and not to be doing it in a specific way all the time, as this information appears in the table and would be redundant. In addition, this makes it easier for the reader to read and understand, so as not to include so many statistical indicators (we insist, they appear in the tables). Thanks again.

We thank you for informing us of the difficulty in understanding the tables, however, following the APA guidelines and taking into account that the journal's guidelines are to establish codes in all the cells of the table, we understand that reading and interpreting them horizontally is appropriate. Thank you very much.

The frequency of devices, understood as how many times a day or week they use them, is not recorded as we consider that controlling this is more complicated and subjective. However, the time of use (expressed in minutes, as specified in the method) of these devices has been recorded. In this way it is specified in the method and is clear so that it is not redundant throughout the text.

Comment

Line 141: correct the sentence, for example, Children residing in Ibero-America spent more time using video consoles, etc. Also, unify the term, Is it Latin America or Ibero-America?

Response

Thank you very much for your comment. Changes have been made throughout the document. We use the term Ibero-America uniformly throughout the document.

Comment

Line 115 to 163: since there are large tables, I suggest writing the name of the table after the data presented, for easier reading and following the results.

Line 189-190: correct the sentence, I suggest similar to Those children whose parents had a basic education level spend longer time using the mobile phone, etc. Il lines afterwards write the time (h or min/week or day) because it is not clear and it is difficult to follow.

Line 198: I suggest not mentioning the European zone, I suggest using the term “compared to the Europeans” or similar. Also, state the name of the value (I suppose that is the time and then provide a unit of it). I suggest not to use “development of school tasks”, perhaps use “school task performance” or similar to this.

Line 203: correct the sentence, rather use “spent more time using video, etc. than “had a long time”. Also, mention the name of the table after the data is presented, and mention “when compared to other education level subgroups”.

Line 253-256: I suggest correcting the sentence, avoid “educational background basics” rather say as it is mentioned in the tables “basic studies” i.e. “basic educational level”.

Response

Thank you very much for all your comments and suggestions. First of all, at the beginning of each paragraph of the results, the reference table for their interpretation is indicated.

Suggested recommendations have been modified and taken into account. Some have not been included because they are not redundant, but apart from changing the terminology, the measures of use have been specified throughout the text (not after each result, as we consider that this is neither precise nor appropriate), as well as being indicated in each paragraph, when a comparison was being made.

Again, we thank you for your effort and time spent in offering your suggestions.

Comment

Line 273: correct the sentence, it is confusing, those are the children residing in Europe, not the parents who spent their time doing PA the most when compared to the children from Latin America.

Response

It has changed.

Comment

Line 275: correct the sentence, those are not the parents, these are the children whose parents had the highest educational level to have the highest time for physical activity. 

Response

It has changed.

Comment

Line 398: correct this sentence, I suggest rephrasing it, not to mention “children in colour” despite the original literature mentioning this way. Be ethical in this issue.

Response

There is no clear answer about this term and it is thought that it has been ethically correct to use this term because a native has told us so. If you know of any other term you can tell us and we will be happy to change it.

Comment

Line 421: this sentence is not clear, is this the meaning of the authors or of the literature authors. Also, correct its original meaning.

Response

It has changed.

Comment

Line 441: there is not clear which one are the reasons, who are they, please, name it, be specific about those reasons and their future implications.

Response

Have been added

Comment

The tables heading are not clear, they should be specific and precise about what is presented. It lacks the parameter's value (time or the score), the number of participants, and their origin place. The tables presented are too long, they could be merged into one according to the same subject. The table could present the mean and SD and a p-value of each parameter in one row and the columns can merge the same subject but for each subgroup (Europe and Latin America). The tables also lack subheadings about which statistical test is applied.

Response

The heading of the tables has been reformulated in an attempt to make it more specific.

Following the guidelines of the journal, the values of the parameters, as well as the number of participants and their origin, have been explained throughout the paragraphs and in the headings of the tables. The editors' recommendations are based on not being redundant with this type of information. However, we are grateful for their suggestion again.

We are aware that some of the tables presented are a bit long, however there are two problems, the first is the template of the journal (which is what it allows) and secondly, we consider that for comprehension and to play around and vary the text and tables, it is easier to understand in this way.

Finally, it should be noted that the statistical tests carried out are reported in the method section. Thank you again for the time and effort invested in reviewing the paper.

Reviewer 2 Report

  What tasks did children between the ages of 3 and 12 spend their time on during the COVID-19 pandemic? An international comparative study between Latin America and Europe

Thank you for the opportunity to review the above entitled manuscript which analyze the degree of psychosocial well-being, sociodemographic aspects and use of technological means depending on the educational level of the parents. 

Information on the impact of social isolation on confinement on children’s life during a pandemic such as SARS-CoV-2/COVID-19 is of public health importance. While I commend the authors for undertaking a study to explore this important issue, I feel there is some information missing from the paper. I have provided specific queries below: 

1.     Introduction - The authors make an introduction based on the effects of COVID-19 Lockdown but, that is already explored in several articles. What I think is the most important here is to explain what measures were implemented in the countries of this MS. What differs them? Because, probably, that differences dictated some of the behaviors founded. Also, is there data on the variables analyzed prior to the pandemic?

2.     Materials and Methods – How where the participants recruited? Which countries where selected and why? Also, I would like to see the survey regarding children daily activities better explained. Did the authors explained the parents the difference between free play, playing with the family and even PA practice? Can free play be considered a form of PA? If the child is playing catch is considered PA or free play? 

3.     Statistic – Although the methods are well applied did the authors considered doing an adjustment for potential confounders such as number of children in the household, type of house, access to outside areas etc?

4.     Discussion – The authors provide an impressive number of references that support their findings but, the MS lacks an explanation of why they encountered these results. And this is difficult to understand. The MS lacks and introduction on the measures taken in the countries analyzed, even the definition of that same countries, and that reflects in the discussion. I believe that the MS would benefit of a review of the discussion addressing the explanation of the findings.

Author Response

After attending to all the comments made by the reviewers, we believe that the work has substantially improved in quality.

Here are the changes made based on your feedback:

Comment

Introduction - The authors make an introduction based on the effects of COVID-19 Lockdown but, that is already explored in several articles. What I think is the most important here is to explain what measures were implemented in the countries of this MS. What differs them? Because, probably, that differences dictated some of the behaviors founded. Also, is there data on the variables analyzed prior to the pandemic?

Response

The theme addressed in this comparison of countries between Europe and Latin America of the tasks carried out by children during confinement is quite new. For this reason, there are few publications that directly address the objective analyzed in this research article. For this reason, it has been difficult to study the measures addressed in each country and it was decided to study it in a more global way.

Yes, there are studies that deal with the habits and tasks that children did before the pandemic. However, it was decided not to address this topic due to the word number and more specifically because the objective analyzed was during confinement.

Comment

Materials and Methods – How where the participants recruited? Which countries where selected and why? Also, I would like to see the survey regarding children daily activities better explained. Did the authors explained the parents the difference between free play, playing with the family and even PA practice? Can free play be considered a form of PA? If the child is playing catch is considered PA or free play? 

Response

Thank you very much for your comments. As stated in point 2.1. the sampling used was by convenience.

In addition, to give more details, a table with the distribution of participating countries has been included.

Being confined, they were told that free play is that game where the child plays with toys, cards, or other objects without the need for a physical activity goal, that is, a goal to improve physical condition. And physical activity was explained to him that it is a game that involves a certain amount of body movement to improve the different physical capacities or motor skills, such as running, jumping, crawling, etc. Playing as a family is another type of game different from free play and activity physical, it was explained to the families that it is a social game where the child interacts with his parents.

Yes, since PA has been considered as any bodily movement that implies a higher metabolic cost than basal, so you can play freely and do physical activity. It would include all those games where the child is not sitting, lying down or standing still.

Yes, because it is producing a higher metabolic cost than at rest.

Comment

Statistic – Although the methods are well applied did the authors considered doing an adjustment for potential confounders such as number of children in the household, type of house, access to outside areas etc?

Response

Thank you very much for your comments and feedback. As we have said, we are grateful for these suggestions, however, this proposal partially departs from the objective of the study, in fact, some of these aspects are reflected in the limitations and will be taken into account for further studies.

Comment

Discussion – The authors provide an impressive number of references that support their findings but, the MS lacks an explanation of why they encountered these results. And this is difficult to understand. The MS lacks and introduction on the measures taken in the countries analyzed, even the definition of that same countries, and that reflects in the discussion. I believe that the MS would benefit of a review of the discussion addressing the explanation of the findings.

Response

Thank you for your comment. The references are very current and are very focused on the theme of the objective indicated in this manuscript.

On the other hand, it is very complex to be able to analyze the measures taken in each of the countries analyzed. For that reason, we have indicated that this is one of the limitations of this manuscript.

Round 2

Reviewer 1 Report

Second review of the manuscript “On what tasks did children between the ages of 3 and 12 years spend their time during the COVID-19 pandemic? An international comparative study between Ibero-America and Europe”

I read the revised manuscript and accepted the authors' responses to the given suggestions and comments. The manuscript has been improved and now is easier to follow. But there are still some parts of the text that need to be checked in English grammar.

Line 295: this sentence has to be corrected for English grammar

Line 329: this sentence is something missing, it should be corrected

Line 337: this sentence has the wrong word “bios”, I suppose it should be “bias”. The authors should check this citation.

Line 400: this sentence should be corrected, and I suggest that the authors compare the citation conclusion with their study results.

Line 405: the authors are suggested to correct this sentence for the Ibero-Americans’ results, I suggest saying “…while it was the Ibero-Americans who had a greater number of televisions.

Line 418: for this sentence, I suggest using “Black children” as the authors of this citation express themselves this way, it is more specific to their study results.

Line 437: this sentence should be corrected, I suggest mentioning “parents” for the exact meaning of their level of schooling and economic situation.

Line 456: correct this sentence to be more specific, I suggest replacing “the highest mean values for all technological devices.” with “a greater number of technological devices.”

Author Response

Thanks for your comments, some English expressions have been modified

Reviewer 2 Report

Thanks for your answers.

Author Response

(The authors gave the same response as above.)
